# Research Evidence of the Role of the Glymphatic System and Its Potential Pharmacological Modulation in Neurodegenerative Diseases

**DOI:** 10.3390/jcm11236964

**Published:** 2022-11-25

**Authors:** Joji Philip Verghese, Alana Terry, Edoardo Rosario de Natale, Marios Politis

**Affiliations:** Neurodegeneration Imaging Group, University of Exeter Medical School, London W12 0BZ, UK

**Keywords:** glymphatics system, neurodegeneration, pharmacological modulation, APQ4, Alzheimer’s disease, Parkinson’s disease, Huntington’s disease, multiple sclerosis, motor neurone disease, idiopathic normal pressure hydrocephalus

## Abstract

The glymphatic system is a unique pathway that utilises end-feet Aquaporin 4 (AQP4) channels within perivascular astrocytes, which is believed to cause cerebrospinal fluid (CSF) inflow into perivascular space (PVS), providing nutrients and waste disposal of the brain parenchyma. It is theorised that the bulk flow of CSF within the PVS removes waste products, soluble proteins, and products of metabolic activity, such as amyloid-β (Aβ). In the experimental model, the glymphatic system is selectively active during slow-wave sleep, and its activity is affected by both sleep dysfunction and deprivation. Dysfunction of the glymphatic system has been proposed as a potential key driver of neurodegeneration. This hypothesis is indirectly supported by the close relationship between neurodegenerative diseases and sleep alterations, frequently occurring years before the clinical diagnosis. Therefore, a detailed characterisation of the function of the glymphatic system in human physiology and disease would shed light on its early stage pathophysiology. The study of the glymphatic system is also critical to identifying means for its pharmacological modulation, which may have the potential for disease modification. This review will critically outline the primary evidence from literature about the dysfunction of the glymphatic system in neurodegeneration and discuss the rationale and current knowledge about pharmacological modulation of the glymphatic system in the animal model and its potential clinical applications in human clinical trials.

## 1. Introduction

The glymphatic system is a waste clearance system that utilises perivascular channels formed by astroglial cells surrounding penetrating arterioles [1,2,3]. Through the transport of cerebrospinal fluid (CSF) through the perivascular space (PVS), the glymphatic system can help eliminate waste products and deliver critical nutrients to the brain [1,2]. It is postulated that CSF inflow into the glymphatic system is regulated by Aquaporin 4 (AQP4) water channels [4], which are highly expressed in the end-feet of astroglia cells that form the PVS [5,6,7]. The exchange of solutes from the CSF and interstitial fluid within the glymphatic system is mainly driven by cardiovascular pulsation [8]. A schematic diagram of the glymphatic flux can be seen in Figure 1. Sleep seems to regulate the function of the glymphatic system, with minimal activation seen during periods when individuals are awake [1,9]. During periods of sleep, expansion of the PVS and increased CSF influx to the glymphatic system helps to remove neurotoxic waste products produced during wakefulness [1,10,11]. Notably, both rodent and human research posit a relationship between deprived sleep and amplified quantities of cerebral amyloid-β (Aβ) [12], a peptide implicated in several neurodegenerative conditions [13]. The efficiency of the glymphatic system also reduces with age [1], with studies suggesting age-related factors such as the loss of perivascular AQP4 polarisation within the microglia [14,15], decreased CSF production [16], reduced CSF pressure [17] and reduced arterial pulsatility [18,19] as main leading causes of such conditions. Age is the most significant risk factor for neurodegenerative diseases, with glymphatic impairment in ageing potentially leading to the aggregation of several proteins that may predispose individuals to develop neurodegenerative pathologies [1,9].

The comprehension of the role of the glymphatic system and its associations with age, sleep and protein clearance could lead to a deeper understanding of a common pathophysiological pathway underpinning several neurodegenerative conditions [9]. Moreover, examination of the glymphatic system is also critical in identifying means of risk-stratification, diagnosis and pharmacological modulation, which may have the potential for disease modification [20]. This review aims to critically outline the primary evidence regarding the dysfunction of the glymphatic system within neurodegeneration. Further, we will discuss the current rationales behind and driving pharmacological modulation of the glymphatic system in the animal model and its potential clinical applications in human clinical trials.

## 2. Glymphatic System and Alzheimer’s Disease

Alzheimer’s disease (AD) is a progressive neurodegenerative condition [21] neuropathologically characterised by the deposition of Aβ plaques and neurofibrillary tau tangles [22]. The phenotype of AD is that of a progressive dementia, dominated by loss of episodic memory, language, visuospatial awareness, and deterioration of higher-order executive processing. Other common symptoms encompass sleep disturbance, changes in mood and personality, and psychiatric disorders [21]. The biological spectrum of AD, whereby neuronal degeneration, deposition of tau, and Aβ can be detected [23], encompasses a wide range of clinical entities such as mild cognitive impairment (MCI) [24]), mild behavioural impairment [25]) and subjective cognitive impairment [26], which may represent the phenotypical underpinning of ongoing neurodegeneration [23]. Understanding the pathophysiology underlying these stages is considered critical as it may represent a target of interest for intervention with the potential to influence the progression of the neurodegenerative process [27].

Exciting possible associations between AD, glymphatic function and sleep have been a contemporary topic of interest [15]. Clinical manifestations of AD can include insomnia and waking hour sleepiness, which have been associated with more severe cognitive deterioration [28]. Notably, sleep is already implicated as a critical element of practical glymphatic function [11] and amplified CSF exchange [10]. Moreover, associations between deprived sleep and raised Aβ have been established [9,12]. However, whether altered sleep is a consequence or driver of disease processes within AD is contentious [15]. A bidirectional relationship linking AD pathophysiology and sleep have also been suggested [29]. Consequently, possible optimisation of the glymphatic system through sleep regulation holds exciting potential as an area for pharmacological [30].

According to the classic Amyloid hypothesis, a key event in the pathogenesis of AD is an imbalance of Aβ manufacture and removal [31]. It has been postulated that the glymphatic system may play a significant role in the extracellular clearance of toxic Aβ aggregates [32]. Furthermore, human PET scanning using [^11^C]-PiB, an Aβ-sensitive tracer proposed to be also sensitive to CSF clearance alterations in AD, has shown defective CSF clearage in AD patients [33]. Regarding the generation of AD pathology, a perpetual cycle of Aβ build-up, glymphatic dysfunction and Aβ accretion may occur, resulting in neuronal demise [34].

The clearance of interstitial cerebral Aβ through the glymphatic system is multifaceted. It involves mechanisms such as glial and neuronal degradation, the crossing of the blood–brain barrier (BBB) and the perivascular movement of interstitial fluid along the glymphatic system towards draining vasculature [34]. Several factors may contribute to the dysfunction of the glymphatic system here. Examples of this include; decreased arterial compliance and consequent disruption of normal pulsatility of the cerebral vasculature (which drives CSF/ISF stream and transfer) as a result of cerebral amyloid angiopathy (CAA) [34,35] as well as disruption of glymphatic flux due to changes in the perivasculature resistance and pressures [35]. Such changes are suggested to be detrimental to the functionality of the glymphatic system due to its posited mechanisms of action. For example, it has been hypothesized that paravascular flux is crucially regulated by diffusion as the principal transportation mechanism [3].

Within the ageing brain, both structural (such as AQP4 perivascular depolarisation) and functional aberration (including decreased intracortical arteriole wall pulsatility) of the glymphatic system and the consequent impairment of paravascular Aβ removal may underpin cognitive decline [14]. This has also been implicated as a causal feature of AD pathophysiology, supported by animal and human research. Some recent studies on the APP/PS1 rodent AD models demonstrated a decrease in glymphatic transportation [36] as well as intraneuronal Apolipoprotein E (ApoE, a lipid transporter within the brain) and Aβ accretion [37]. Here, relative to APP/PS1 groups, AQP4-/-APP/PS1 rodents demonstrated an increase in cerebral cortex microglial Aβ activation and phagocytosis [37].

As previously mentioned, AQP4 is speculated to play a role in the pathological aggregation of Aβ. AQP4 is a water channel highly polarised in the astrocytic end-feet fringing perivasculature basal membranes [34] that regulates tonicity-responsive cross-membrane transport of water [38]. In physiological conditions, AQP4 is crucial for efficacious glymphatic movement and CSF clearing [39,40,41] via transastrocytic bulk flow [42], as the majority are localised so that APQ4 are connected to adjacent perivascular spaces, enabling CSF flow between this and the cerebral parenchyma [15]. Here, the admixture of CSF and interstitial fluid (ISF) occurs, which then travels to subarachnoid spaces where the CSF-ISF then infiltrates into the lymphatic or bloodstream vessels [42]. As a result, this facilitates a drainage/clearance mechanism of the ISF and, notably, the removal of waste products such as Aβ [43]. Therefore, in a pathological setting, AQP4 is hypothesised to be implicated in Aβ clearance [5]. This has been supported by studies on APQ4 knock out mice showing a decreased removal of solutes in the interstitial space and a decelerating influx of CSF [5]. Moreover, clearance of fluoro-labelled Aβ was also inhibited, suggesting a causal correlation between AQP4 loss and decreased removal of solutes [5].

The exact role of AQP4 dysfunction in AD is not fully understood or established. However, hypotheses have been put forth. It has been noted that in both animal AD models and human AD post-mortem tissues, there is a depolarisation of astrocytic AQP4 [44] and loss of localisation of AQP4 along the astrocytic end-feet relative to healthy controls [45]. Due to this loss of AQP4 polarisation, the glymphatic function is impaired, with reduced toxic metabolite and CSF clearance [46]. Furthermore, APP/PS1 mice models carrying an *APQ4* genetic deletion (AQP4^−^) demonstrated CAA and augmented build-up of Aβ, leading to the postulation of the attenuating impact of AQP4 in relation to Aβ and this as a promising potential target in future AD therapeutic intervention [47]. Additionally it has been demonstrated in human studies, that polymorphisms in AQP4 can predict Aβ uptake and clinical outcomes in patients with AD or late-MCI [48], however this association remains contentious [15]. For example, there is debate as to whether AQP4 depolarisation is the causative driver of (and therefore precedes) a decrease in Aβ clearing [45] or if rather the former is consequential of the latter [44]. Therefore, it is apparent that further research is needed to establish more conclusive evidence regarding this [15]. A summary of findings from an investigation of the glymphatic system in both human and animal AD models is shown in Table 1.

## 3. Glymphatic System and Parkinson’s Disease

Parkinson’s Disease (PD) is a neurodegenerative disorder characterised by the aggregation of α-synuclein in cellular aggregates named Lewy-Bodies (LB), putatively leading to progressive damage and loss of neurons in the brain [49,50]. Classical motor symptoms encompass bradykinesia, resting tremor and rigidity [51,52] and have been attributed mainly to dopaminergic neuronal loss in the substantia nigra. Although the build-up of α-synuclein is believed to be associated with PD pathogenesis, other factors, such as sleep disturbance, are associated with disease progression [53]. Epidemiological studies have long identified sleep disturbance or working night shifts as associated with a higher frequency of developing PD later in life [54,55]. REM sleep behaviour disorder (RBD) is defined by the loss of muscle atonia during REM sleep, causing patients to act out their dreams [56]. RBD has been identified as a prodromal manifestation of synucleinopathies, and a precursor to clinical PD [57], with studies identifying that 20–77% of PD patients have symptoms of RBD before the onset of motor symptoms [58,59,60,61]. Interestingly, sleep disturbances in PD have been associated with a particular PD phenotype. RBD in PD patients is associated with more severe motor and non-motor symptoms [62], and increased sleep disturbance is associated with increased motor impairment in PD subjects [63].

Increasing evidence suggests that AQP4 plays a critical role in the pathophysiology of PD. Studies on APQ4—mice with experimental PD have shown reduced anti-inflammatory activity levels as evidenced by reduced transforming growth factor-β1 levels along with reduced CD4+ and CD25+ regulatory T-Cells levels [64,65,66] with increased pro-inflammatory pathway activity and cytokines levels, as demonstrated by the increase of the levels of TNF-α, IL-1 and IL-1β, with a rise in the activity of NF-κB [65,67].In addition, in two studies that administrated 1-methyl-4-phenyl-1,2,3,6-tetrahydropyridine (MPTP), a toxin that selectively damages the nigrostriatal dopaminergic system, AQP4- mice were at significantly greater risk of neurotoxicity, as shown with increased inflammatory response and greater dopaminergic neuronal loss, when compared to AQP4 wild-type mice [66,67]. Alongside its potential driver effect on neuroinflammation, in the animal model, reduced AQP4 expression has also been associated with greater α-synuclein deposition and progressive dopaminergic neuronal loss in the SN [68]. Some hints of a similar mechanism taking place also in human disease come from a *post-mortem* study on brain tissue from patients with PD [69]. In this work, a negative correlation between regional detection of AQP4- and AQP1-positive astrocytes and local α-synuclein deposition in neuronal layers II-III and V-VI was seen, suggesting that the excess deposition of α-synuclein could be facilitated by a local environment depleted of AQP4 and AQP1- expressing astrocytes [69].

Aβ retention can also be present in a minority of PD patients [70]. A recent 3-year longitudinal study [71] found that PD patients with enlarged Basal Ganglia PVS (BG-PVS) at baseline had reduced CSF Aβ_42_ and had lower Montreal cognitive assessment (MoCA) scores at follow-up compared to PD patients without enlarged BG-PVS. However, several studies have shown a low prevalence of Aβ with PD without dementia [72] and those with MCI [70]. In a study utilising Positron Emission Tomography (PET) imaging with [^18^F]Florbetaben in a large cohort of PD subjects (ranging from PD patients with normal cognition to subjects with MCI to Parkinson’s Disease Dementia), increasing age was associated with increased cortical [^18^F]Florbetaben retention in the PD group. However, after adjusting for age, no correlation was found between cortical [^18^F]Florbetaben uptake and global cognitive ability within the PD cohort [73]. Results from these studies suggest that Aβ accumulation is not the primary cause of cognitive impairment in PD, rather the presence of Aβ synergises with the other pathological processes, which can accelerate primary cognitive impairment seen in PD [70,73]. At the time of this review, no specific studies have explored Aβ changes specifically within the glymphatic system of PD patients. A summary of metabolite and AQP4 changes in PD is shown in Table 2.

## 4. Glymphatic System and Huntington’s Disease

Huntington’s disease (HD) is a devastating autosomal dominant neurodegenerative condition, clinically manifesting with a combination of motor, neuropsychiatric and cognitive symptoms [80,81]. HD is caused by the pathological expansion of a CAG trinucleotide repetition in the *HTT* gene, resulting in the production of mutant huntingtin (mHTT) [82]. There is currently no curative agent available for HD [83].

Whilst the currently available research base remains relatively limited, studies have suggested that disruption of the glymphatic system may play a role within HD [9]. For example, disrupted sleep is known to occur commonly in HD [84], and the study of the relationship between altered sleep and the function of the glymphatic system in early premanifest carriers of HD mutations would represent a fruitful field of research in this pathology [9].

In HD, a correlation between clinical metrics with the levels of CSF mHTT has been established [85,86]. Some studies have further examined cerebral mHTT removal mechanisms; it has been shown that the latter is impacted by cellular mHTT secretions and then extracellular removal via the glymphatic system [85]. Recent work examined mHTT elimination processes in vivo and in vitro rodent neuronal and astrocyte secretion studies [85]. To further establish glymphatic influence on the cerebral HTT clearance to CSF, central nervous system *AQP4* suppression was performed using antisense oligonucleotides. Notably, this resulted in reduced mHTT in plasma and CSF, implicating the contribution of an AQP4-mediated process, possibly implying glymphatic clearance mechanisms, in extracellular cerebral mHTT removal. Possible implications of such findings include the utility of such measures as clinical trial markers [85]. Further evidence comes from another work in HD model rodents, in which intraparenchymal delivery of antisense oligonucleotides injected following cisternal injection was significantly lower in AQP4-/- mice as opposed to the wild-type mice, suggesting that modulation of the glymphatic system could also influence the optimal delivery of drugs for this condition (Wu et al. 2020). It is therefore evident that, whilst still in the early stages of emerging, modulation of the glymphatic system may provide novel contributions to HD management

## 5. Glymphatic System and Motor Neuron Disease

Motor Neuron Disease (MND) is a disease entity in which neurodegeneration selectively affects motor neurons [87]. Amyotrophic lateral sclerosis (ALS) is the most common form of MND; it is characterised by the loss of both upper and lower motor neurons in the motor cortex, brainstem and spinal cord, leading to persistent and progressive weakness [87]. Around 90% of ALS cases are sporadic, with around 10% being familial cases [88].

Pathological changes to AQP4 have been hypothesised to play a role in the pathophysiology of ALS. Overexpression of AQP4 has been observed within the spinal cord, brainstem, and the motor cortex in rodent ALS models [89,90,91]. In a study utilising mice with superoxide dismutase 1 (*SOD1*) G93A mutation (*SOD1^G93A^*), the expression of AQP4 in the spinal cord was increased together with disease progression, but this was coupled with a reduction of AQP4 polarity within the end-feet of astrocytes in the spinal ventral horn in *SOD1^G93A^* mice, in both early and late stages of ALS, with further analysis identifying down-regulation glutamate transporter-1 (GLT-1) in the end-stage *SOD1^G93A^* mice [91]. This could be explained by hypothesising that, in ALS, AQP4 depolarisation may lead to motor neuron dysfunction via the reduction of GLT-1 expression [91]. Reduction of BBB integrity and increased permeability caused by an AQP4-mediated impairment of potassium and connexin regulation is another potential mechanism of brain pathology mediated by AQP4 in ALS [92,93].

The exact role of AQP4 in regulating the BBB and neurovascular units has not truly been established yet; with variable changes in BBB permeability and disease outcomes in different AQP4^−^ animal disease models [94,95,96,97,98]. One study found that *SOD1^G93A^* AQP4^−/−^ ALS mouse models had improved BBB permeability compared to AQP4^+^ *SOD1^G93A^* mouse models, with reduced hemosiderin deposition and immunoglobulin leakage; however, AQP4^−^ *SOD1^G93A^* mice showed an earlier age of disease onset and shorter lifespan compared to AQP4^+^ *SOD1^G93A^* mice [99]. The authors of this study suggested three potential causes for these findings; firstly, AQP4 deficiency reduces outcomes due to cytotoxic oedema but worsens outcomes due to vasogenic oedema. Secondly, disruption of BBB permeability impairs the entry of neurotoxic and neuroprotective substances. The theory suggests that AQP4 deficiency leads to impaired glymphatic functions, leading to waste product build-up and accelerated disease progression in the AQP4^−^ group [99]. The propagation of ALS may not be due to one specific pathway but rather a multifocal mechanism which leads to disease onset and progression. Understanding changes within the glymphatic system may provide critical information to better understanding the unknown elements of ALS whilst improving the understanding of the pathological changes seen with disease onset and progression. A summary of BBB, AQP4 and glymphatic changes in ALS has been shown in Table 3.

## 6. Glymphatic System and Idiopathic Normal Pressure Hydrocephalus

Idiopathic normal pressure hydrocephalus (iNPH) is a neurodegenerative condition [100] characterised by a pathognomonic triad of urinary incontinence, impaired gait and dementia [101]. iNPH is most prevalent in elderly patients, and thus with an ageing population, it is predicted there will be an exponential rise in future incidence rate [102]. Therefore, deepening the understanding of the underlying pathophysiological mechanisms within iNPH is of great interest [103].

The exact aetiology of iNPH has yet to be fully understood [104]. It has recently been suggested that the glymphatic system may be implicated in iNPH pathogenesis [105], with studies showing a significant decrease in glymphatic clearance efficiency for subjects with iNPH relative to healthy controls [105,106]. Notably, a decrease in the glymphatic clearing was noted within the entorhinal cortex (ERC), implicating the ERC’s deterioration as a herald of dementia that manifests in iNPH [106].

Furthermore, reduced expression of AQP4 has also been shown in cortical biopsies from human iNPH subjects [107,108]. It is postulated that this leads to pathology by disrupting glymphatic homeostasis [109]. Moreover, consequential ischaemia may result from cerebral matter displacement, resulting from CSF imbalances [103,109]. Here, hypoxic conditions are detrimental to tissue metabolism and contribute to neurotoxin build-up and neurodegeneration [110]. Glymphatic system mechanisms may also be impacted by BBB leaking of proteins and aberrant CSF circulation [103].

From this brief overview, it is evident that exploring the role of the glymphatic system in iNPH pathogenesis may help understand its aetiology and subsequent prevention and management [103]. Here, the glymphatic system has also been suggested to have future clinical potential, with some research noting dynamic variations in biomarkers within the CSF of iNPH subjects [111], although this notion remains very novel and currently unsubstantiated [103]. Improved diagnosis and thus optimised treatment intervention can alleviate the biopsychosocial implications that may otherwise incur as a result of iNPH incidence [34].

## 7. Glymphatic System and Multiple Sclerosis:

Multiple Sclerosis (MS) is a demyelinating, autoimmune disease of the CNS. Repeated neuroinflammatory bouts lead to the formation of plaques, which can progress to the loss of oligodendrocytes and other neurons [112]. Symptoms of MS are highly variable, ranging from sensorimotor, cerebellar, visual, gastrointestinal and genitourinary symptoms [113]. Due to the presentation and progression of symptoms, MS has a variety of subtypes ranging from relapsing-remitting MS (RRMS), secondary progressive MS, primary progressive MS, and progressive-relapsing MS [113].

Experimental autoimmune encephalitis (EAE) is commonly induced in animal models to emulate demyelinating inflammatory disorders. A recent study aimed to compare AQP4 expression changes in metabolic (cuprizone toxin-induced demyelination) and autoimmune myelin injury (cuprizone + EAE) mice models and then tested in *post-mortem* tissue from progressive MS patients [114]. This study identified diffuse AQP4 expression increases in chronic-active lesions in the advanced MS cohort, with increased anti-AQP4 immunoreactivity observed in immunohistological slides. Increased AQP4 expression has also been seen in studies exploring human brain tissue of patients with an infraction [115] and other inflammatory conditions [116], with the authors of this study suggesting the overexpression of AQP4 within chronic MS lesions is not a specific marker for MS, but rather a broader marker of gliosis and astrocyte pathology. Loss of AQP4 polarity within the end-feet of astrocytes surrounding perivascular structures at the perivascular end-feet of astrocytes and diffuse AQP4 expression increases were seen in the cuprizone toxin-induced demyelination mouse model group [114]. Diffuse increases in AQP4 expression were seen in chronic MS lesions of post-mortem human subjects and cuprizone-induced demyelination models; with authors theorising that increased AQP4 expression was a result of metabolic injury to the brain in both MS subjects and demyelination animal models. The cuprizone positive EAE mice showed increased AQP4 expression at the centre of the inflammatory lesions with reduced AQP4 expression and polarity at the edge of the lesion, with the authors theorising perivascular AQP4 loss is a result of immune cells penetrating the normal brain parenchyma during the acute immune-mediated response [114]. Results from the study show that cuprizone models could potentially transpose to chronic MS lesions in humans. However, further research is needed to determine this. Supposing a definitive model that can emulate the AQP4 changes in acute MS lesions is identified, AQP4 modulating interventions may be assessed to determine their impact on acute MS flares and subsequent progression to chronic lesions. Results from the study above [114] correlate with post-mortem MRI studies, which have identified enlarged PVS (EPVS) in focal and diffuse white matter lesions in MS patients, with histological analysis revealing leukocyte infiltration within the EPVS [117].

In a recent retrospective cross-sectional study on human MS patients [118], the diffusion tensor imaging along the perivascular space (DTI-ALPS) index was found to be significantly lower in both RRMS and progressive multiple sclerosis patients compared to HC, with progressive multiple sclerosis patients exhibiting lower ALPS values than RRMS patients. A lower ALPS index score in the MS groups was associated with more severe clinical disability, more significant lesion load, grey matter atrophy, reduced mean diffusivity, and fractional anisotropy in normal-appearing white matter [118]. Further analysis revealed a negative association between the ALPS index and disease duration in the first 4-years of MS, with no other association. The results of this study indicate that impaired glymphatic function may lead to the accumulation of neuroinflammatory and neuro-toxic factors, resulting in progressive demyelination and neuronal loss.

A summary of AQP4 and glymphatic changes in MS is shown in Table 4.

## 8. Glymphatic System and Traumatic Brain Injury (TBI)

Traumatic brain injury (TBI) can be described as an indication of cerebral pathology, brain injury and altered brain function due to external force, such as explosive, closed or penetrating head trauma [119]. TBI pathophysiology is multifaceted and not yet completely understood [120] but is considered to involve and be related to the interplay of primary and secondary trauma reactions (e.g., metabolic alterations, neuroinflammatory processes and BBB disturbance) [121].

TBI studies have observed tau phosphorylation and Aβ deposition in rodent and human TBI studies [122,123,124]. As such, the glymphatic system and its role in waste clearance have been suggested to play a role in TBI [125]. In mice studies, there was approximately a 60% decrease in glymphatic system functionality following TBI, which persevered for a minimum of a month following the insult, inciting tau accumulation [126]. A further contrast-enhancement-MRI study on mild TBI rats showed a significantly decreased contrast clearance rate, suggesting glymphatic inefficiency [127]. Moreover, AQP4 re-localisation has also been noted in the acute post-TBI stage, which may negatively impact waste clearance [128]. For example, APQ- post-TBI rodents demonstrated exacerbated tau accumulation [129].

It is evident that further investigation of the role of the glymphatic system in TBI is necessary and provides an exciting basis for future research aiming to establish clinical utility/application [130].

## 9. Pharmacological Modulation of the Glymphatic System in Neurodegenerative Diseases

Novel research is starting to present the rationale for the pharmacological modulation of the glymphatic system [1,131] for therapeutic purposes. As discussed, animal models have shown that AQP4 deletion is related to increased accumulation of proteins such as Aβ and tau within the brain [37,132]. Several studies have theorised the use of AQP4 therapies in several conditions, such as stroke and AD; but no specific human trials utilising AQP4 treatments for these have yet to be conducted from the date of this review and the exact mechanism of this remain complex and not fully comprehended [20,133,134].

However, AQP4 modulation and how this may alter the levels of proteins associated with neurodegeneration remains an exciting topic, despite remaining relatively novel [15,46]. As aforementioned, whilst the current research base remains limited, some studies have already shown promising effects from AQP4 modulation, including AQP4 inhibition.

Initial pre-clinical research tested TGN-073 (N-(3-benzyloxypyridin-2-yl)-benzene-sulfonamide), an innovative facilitator of AQP4, on mice models. Using labeled water, [^17^O]H2O JJ vicinal coupling proton exchange MRI, they found increased turnover of labeled water within the PVS in TGN-073 treated mice [135]. Thus, it was further implicated in vivo that AQP4 plays an important function in interstitial circulation, but furthermore that pharmacological facilitation of APQ4 activity is an exciting avenue for the potential management of conditions such as AD [135].

In turn, TGN-073 structural congeners, including small molecule modulators such as N-(1,3,4-thiadiazol-2-yl) pyridine-3-carboxamide dihydrochloride (TGN-020) have since been demonstrated to be AQP4 inhibitors using ubiquitin-proteasome systems intracellularly. TGN-020 is already established to inhibit AQP4, with an inhibitory IC_50_ value of 3 [136]. For example, in non-reperfusion ischemic (via occluding medial cerebral arteries) rodent models, inhibition of AQP4 with TGN-020 has shown a decrease in glial scarring, apoptosis and cerebral edema [79,137], thus implicating the potential of such interventions in modulating clinical outcomes of pathologies such as ischemic stroke [46]. Moreover, a negative correlation between polarization of AQP4 and infarct region astrogliosis was noted, leading to the postulation that preservation of AQP4 polarization (which furthermore showed a positive relationship with diffusion weight imaging apparent diffusion coefficients) could be achieved via interventions addressing astrogliosis [79]. Similar reductions in osmotic flux have been demonstrated using high concentration bumetanide, a loop diuretic. Here, via the blockage of Na(+)-K(+)-2Cl(-) co-transport and AQP4 inhibition, a decrease in cerebral edema in post-stroke rodents was demonstrated, suggesting another potential pharmacological approach to conditions whereby fluid imbalances or edematous changes manifest [138].

Several other agents have furthermore been shown to inhibit AQP4 such as antiepileptic drugs [139]. Such findings stemmed from evidences in APQ4 knock out rodents demonstrating remonstrance to pentylenetetrazol, a chemical convulsant. This in turn lead to the hypothesis that the action of antiepileptic drugs may be, at least somewhat related to an impact on AQP4 [139]. Arylsulfonamides [140], bumetanide, and quaternary ammonium compounds [141] have also shown similar inhibitory effects.

Moreover, another study demonstrated effective AQP4 modulation using SR49059 which selectively antagonizes vasopressin V1a receptors. Arginine Vasopressin (AVP) is an important factor contributing to the pathophysiology of cerebral edema following, for example, ischemic brain injury [142]. There is a temporal relationship between the increase of vasopressin and the decrease of GFAP and AQP4-mediated astrocytic function so to suggest a causative association [143]. This concept has now expanded to include other conditions where a dysfunction of the glymphatic system has been demonstrated, such as TBI [144] Inhibition of AVP may therefore seem a promising method to improve the functionality of the glymphatic system through modulation of AQP4. For example, SR49059 is a strong antagonist of V1 vasopressor [145]. Taya and colleagues demonstrated, in a rat model of TBI, that SR49059 reduced AQP4 levels and, consequently, brain water content, with a general beneficial effect on the trauma-induced brain injury [146]. This result was confirmed by another work [147], therefore suggesting that inhibitors of vasopressin could represent therapeutic tools to modulate the function of the glymphatic system, with potential effect also outside cerebrovascular disease.

Another pharmacological agent of interest within this discussion is thyroid hormone 3,3′,5-triiodo-l-thyronine (T3). There is an increasing body of evidence to suggest the role of thyroid function within neurological pathologies [148]. It is already established that such hormones are implicit in cerebral developmental growth, and moreover, there has been associations between decreased T3 and more adverse clinical outcomes following stroke [149]. Interestingly, reduced AQP4 expression and anti-edematous effects have been shown in ischemic stroke rodents following T3 administration [150]. Here, it is suggested that such findings and anti-edematous results may be a consequence of suppression of AQP4 expression, providing a foundation for further research into the use of T3 within ischemic neurological conditions [150].

It is however important to note that whether such changes may be translated into clinical application is still unclear [131]. Similarly, the possibility that the AQP4 changes discussed may be a result of indirect mechanisms, as opposed to direct binding must be considered [151]. Furthering the understanding of AQP4 modulation thus holds great promise for future pharmacological application to the glymphatics system. A summary of the potential independent AQP4 pharmacological regulation of the glymphatics system discussed above can be seen in Figure 2.

Exploiting the previously discussed relationship between sleep and glymphatic function is another exciting avenue of potential [11]. Here, with regards to pharmacological modulation, the role of melatonin is of particular interest [152,153]. Melatonin is a pineal hormone that regulates circadian rhythms (i.e., sleep-wake cycles) [154]. In AD animal studies, research has shown promising findings with melatonin treatment reducing Aβ within cerebral mitochondria [152]. Increased Aβ clearance has also been demonstrated in melatonin-treated transgenic rodent AD amyloidosis models [155], thus implicating its possible role in ameliorating glymphatic clearance [153]. Whilst the specific pharmacological understanding underpinning such findings remains under investigation, this highlights a valuable starting point for further research into pharmacological modulation of the glymphatic system [156].

A further potential avenue for the pharmacological modulation of the glymphatic system is the adrenergic system. The rationale for this comes from in vivo 2-photon imaging of rodents whereby those sleeping and anesthetised showed substantially increased CSF influx, suggesting increased glymphatic activity, which was not present in those awake [1]. Further CSF tracer distribution has been examined following anaesthesia. For example, following Ketamine/Xylazine anaesthetic administration in mice, tracer influx in CSF and, notably, efflux of radio- Aβ was analogous to levels found in natural sleep states [157]. Enhanced glymphatic functionality has also been demonstrated with Propofol and Phenobarbital usage in mice [156,157]. Agents such as Xylazine are potent agonists of α_2_-adrenergic receptors, reducing norepinephrine release, a principal modulator of arousal [158]. This has been corroborated by the findings of near sleep/anaesthesia state equivalent increases of tracer flux within the CSF in wakeful rodents post-administration of norepinephrine receptor antagonists [10]. Thus, Norepinephrine has been implicated in playing a critical regulatory role in glymphatic function. Studies show an increased clearance rate due to extracellular space expansion and reduced resistance due to lowered Norepinephrine amounts in a sleep state [159]. Moreover, local delivery of other antagonists (including Prazosin (which is antagonistic of alpha 1 adrenergic receptors), Propranolol (an antagonist of β-adrenergic receptors) and Atipamezole (an alpha-2 adrenergic antagonist)) in animal studies extracellularly demonstrated similar findings [160]. Furthermore, such adrenergic modulation has already been shown to improve the quality of sleep in TBI and PTSD patient demographics, thus the potential extrapolation of this effect within a wider population is of great interest [161]. Such results provide an excellent foundation for future studies investigating glymphatic function [161]. Further exploitation of the glymphatic system to facilitate the delivery of pharmaceuticals from the CSF to the cerebral parenchyma is another promising area of interest [162]. Research in this field has included investigating the effect of glymphatic modulation using Dexmedetomidine, an anti-anxiolytic and analgesic medication which also has sedative effects that bind to α_2_-adrenergic receptors with high selectivity) [163]. Here altered pharmacokinetics within the CNS (increased cerebral exposure) of systemic or intrathecal drugs was shown [162] Thus, it was postulated that glymphatic modulation enabled an augmented CNS dispersal of low molecular weight pharmaceuticals [162], raising the postulation as to whether it could act as a glymphatic enhancer [164].

Further research has also established an association between glymphatic dysfunction and slow wave activity (SWA) [165] [166], with an inverse correlation noted between SWA and amounts of the pathological AD markers tau and Aβ [167]. This, in turn, may demonstrate the potential utility of SWA modulation as a novel therapeutic target in conditions such as AD [165]. Working off this notion, several pharmacological agents have already been shown to enhance SWA, including Sodium Oxybate [168], Olanzapine, Tiagabine and Baclofen [165]. It is theorised that this modulation and augmentation of SWA may provide a novel non-invasive therapeutic method of enhancing cerebral Aβ clearance in treating conditions such as AD in the future [169].

Adenosine is a somnogenic neuromodulator that plays a role in sleep-wake regulation [170], shown to increase in the sleep-deprived basal forebrain [171]. Moreover, it has been demonstrated that in TBI, dysfunction of perivascular AQP4 can be ameliorated via the inactivation of adenosine A_2A_ receptors [172]. Substances such as caffeine are antagonists of A1/A2A adenosine receptors [173]; therefore, extrapolating such mechanisms may benefit future glymphatic research. 5-caffeoylquinic acid (a component of tea and coffee) administration in AD rodents has decreased Aβ accumulation through amplified glymphatic clearance [15]. Such findings reinforce the grounds for the prospect of potential future modulation and targeting of the adenosinergic system in relation to glymphatic dysfunction in neurodegeneration [172].

Another upcoming approach of potential utility in glymphatic modulation is photobiomodulation (PBM) therapy, which is defined as the utilisation of light (red or near-infrared) to elicit advantageous changes in cellular biochemistry [174]. Contemporary animal research has explored the impact of PBM on glymphatic clearance [175]. For example, a rodent pilot study found significantly decreased Aβ deposits following transcranial PBM and improved memory and cognition in AD models [175]. Other studies have also demonstrated vasodilation of lymphatic vasculature and augmented lymphatic drainage with low-dose PBM [176]. Thus, it has been implicated that PBM therapy targeting the glymphatic system may have great potential to enhance flux clearance and drainage and offer a promising therapeutic approach for neurodegenerative conditions where glymphatic drainage is aberrant [177]. It is evident, however, that extensive further research (i.e., investigating physiochemically disparate drugs) is required to elucidate this premise and further establish the glymphatic system’s speculative prospects in this regard [162]. Whilst such findings provide an exciting potential for future research, it is evident that a greater understanding of the complex biological mechanisms of AQP4 and the glymphatics system is first required [46]. Similarly, for many neurodegenerative pathologies, pharmacological modulators of AQP4 appear not yet to have commenced clinical trialing stages or remain pre-clinical [15].

## 10. Conclusions and Future Directions

In conclusion, the dysfunction of the glymphatic system plays an important, multifaceted, yet largely mysterious role in several neurodegenerative diseases. For example, it is uncertain if the glymphatic system changes play a primary role in the pathogenesis of disease onset and propagation or a secondary response to changes associated with neurodegenerative conditions. In such diseases, exploring the available evidence on the modulation of the glymphatic system presents a highly fascinating, promising and novel avenue for future research and potential clinical applications in human clinical trials [178]. It remains clear, however, that extensive further study (such as longitudinal research) is required to clarify the glymphatic system’s exact role in neurodegenerative pathologies and how we may effectively modulate this to achieve biological effects in neurodegenerative diseases.

## Figures and Tables

**Figure 1 jcm-11-06964-f001:**
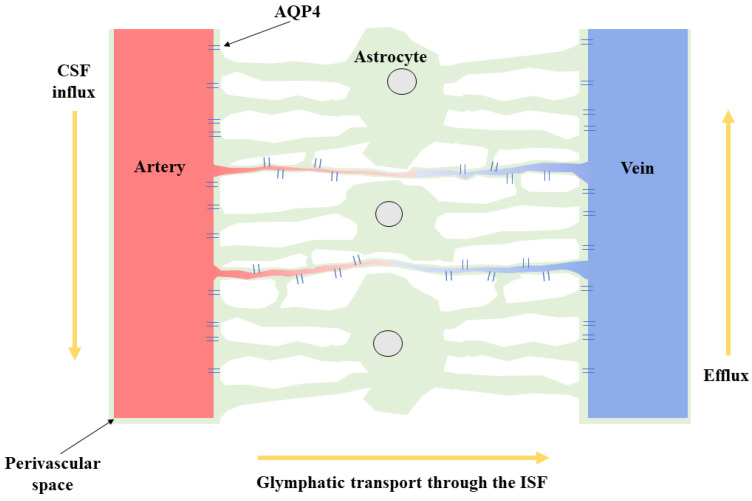
*Glymphatic transport through the interstitial fluid (ISF).* Nutrient transfer and waste clearance occur via the glymphatic system by utilising perivascular channels established by astroglia surrounding supplying. Here, the image depicts the vessels and surrounding aquaporin-4 (AQP4) water channels, that are highly expressed on astroglial end-feet, which regulate cerebrospinal fluid (CSF) flux in the direction of the arrow to be returned to the venous system. The exact mechanisms of this transport remain contentious (i.e., via cardiovascular pulsation or diffusion etc).

**Figure 2 jcm-11-06964-f002:**
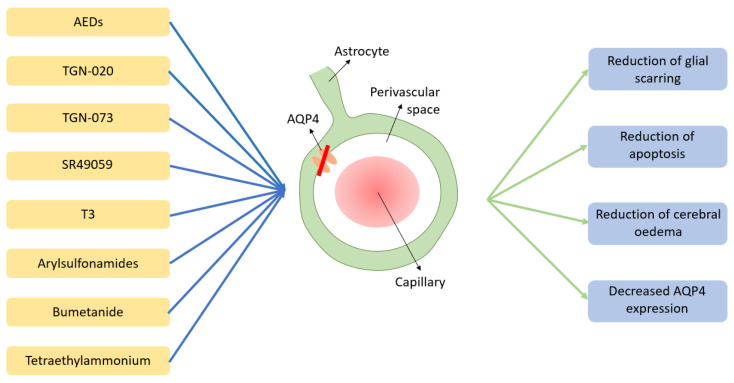
*Potential/speculative pharmacological modulators of the glymphatic system and their potential mechanism of action.* Pharmacological agents shown in yellow boxes all exert a modulating effect on AQP4, resulting in a net inhibition (red line) of its activity. The blue boxes outline the suggested biological effects of AQP4 modulation as emerged from available literature. Abbreviations: AEDs = anti-epileptic drugs, AQP4 = Aquaporin 4; TGN-20 = 2-(nicotinamide)-1,3,4-thiadiazole, TGN-073 = (N-(3-benzyloxypyridin-2-yl)-benzene-sulfonamide), T3 = 3,3′,5-triiodo-l-thyronine.

**Table 1 jcm-11-06964-t001:** Summary of findings of investigation of the glymphatic system in both human and animal AD models.

Cohort	Main Findings	References
100 LMCI or mild AD (Aβ +ve), 469 MCI (168 Aβ +ve, 145 Aβ -ve), 244 LMCI, 97 Aβ -ve CN controls	AQP4 SNP rs72878794 = ↓ uptake of Aβ AQP4 SNP rs151244 = ↑ uptake of Aβ	(Chandra et al. 2021) [48]
APP695/PS1-dE9 transgenic (APP/PS1), AQP4−/−/APP/PS1, WT, APQ4 KO (AQP4−/−) rodents (3 mnth)	Relative to APP/PS1 groups, AQP4-/-APP/PS1 rodents = ↑ cerebral cortex microglial Aβ activation and phagocytosis	(Feng et al. 2020) [37]
C57BL/6 rodents (8–12 wks M)	↓ CSF influx in AQP4- rodents	(Iliff et al. 2012) [5]
Rodents [M + F, APPswe/PS1dE9 (APP/PS1), C57BL/6J, Tg (Cspg4-Ds Red.T1)1Akik/J (NG2-DsRed reporter mice), LM controls	APP/PS1 = Aβ accumulation, ↓ glymphatic clearance	(Peng et al. 2016) [36]
11 AD 12 MCI 20 MS Matched controls	[^11^C]-PiB PET = ↓ signal clearance (lat ventricles) AD vs. HC.	(Schubert et al. 2019) [33]
APP/PS1 mice (12-mnths)	AQP4- APP/PS1 = ↑ astrocyte atrophy, CAA, Aβ build-up, ↓ cognition	(Xu et al. 2015) [47]
79 total PM (cog intact 33–57yrs, Cog intact 61–96, AD 60+)	Assoc. between AD and localization of AQP4 (PV) Assoc. with AQP4 (PV) localization and ↑ Aβ	(Zeppenfeld et al. 2017) [45]

Abbreviations: +ve = positive, ↑ = increased, ↓ = decreased, AD = Alzheimer’s disease, AQP4 = aquaporin 4, assoc. = association, Aβ = amyloid, CAA = cerebral amyloid angiopathy, CN = cognitively normal, cog = cognitively, CSF = cerebrospinal fluid, F = female, HC = healthy controls, KO = knock out, lat = lateral, LM = littermate, LMCI = late mild cognitive impairment, M = male, MCI = mild cognitive impairment, mnths = months, MS = multiple sclerosis, PM = post mortem, PV = perivascular, SNP = single nucleotide polymorphism, -ve = negative, wks = weeks, WT = wild type, yrs = years.

**Table 2 jcm-11-06964-t002:** Summary of the main findings from studies exploring glymphatic changes in Parkinson’s Disease.

Cohort	Main Findings		References
PD: Mouse models	Inflammatory changes	AQP4−/− mice models showed reduced anti-inflammatory activity levels (reduced transforming growth factor-β1 levels along with reduced CD4+ and CD25+ regulatory T-Cells levels) compared to AQP4+/+ mice models.	(Chi et al. 2011; Sun et al. 2016; Xue et al. 2019) [64,65,66]
		AQP4−/− mice models showed increased pro-inflammatory pathway activity and cytokines levels (increased TNF-α, IL-1 and IL-1β levels with increased NF-κB activity) compared to AQP4+/+ mice models.	(Sun et al. 2016; Zhang et al. 2016) [65,67]
	Inflammatory changes after MPTP changes	AQP4−/− mice models were at significantly greater risk of MPTP neurotoxicity compared to mice with AQP4+/+ mice models.	(Zhang et al. 2016) [67]
		AQP4−/− mice models had significantly greater pro-inflammatory markers after MPTP delivery than AQP4+/+ mice models.	(Zhang et al. 2016) [74]
		AQP4−/− mice models had significantly lower transforming growth factor- β1 (a suppressive cytokine) after MPTP delivery than AQP4+/+ mice models.	(Xue et al. 2019) [66]
	α-synuclein	Reduced AQP4 expression was associated with greater α-synuclein deposition and progressive dopaminergic neurone loss within the SN (when comparing AQP4^−/−^ and AQP4^+/+^ mice models)	[68]
PD: Human	α-synuclein	Greater expression of AQP4 and AQP1 positive astrocytes were seen in the temporal lobe of the neocortical PD group compared to HC as well limbic and brain stem PD subgroups.	(Hoshi et al. 2017) [69]
		PD cohort revealing a negative correlation between AQP1/AQP4 to α-synuclein deposition in neuronal layers II-III and V-VI respectively.	(Hoshi et al. 2017) [74]
	ApoE	APOE rs405509 T allele is correlated with increased susceptibility of PD in a Chinese population.	(Huang et al. 2020) [75]
		Early PD patients with ApoE ε4 allele mutations have shown a more rate rapid cognitive decline than early PD patients without ApoE ε4 allele mutations.	(Kim et al. 2021) [76]
		Non-demented PD patients with ApoE ε4 allele mutations have shown worse cognitive performance scores than non-demented PD patients without ApoE ε4 allele mutations.	(Tipton et al. 2021) [77]
		Increased LRP1 and ApoE in LB and melanised neurons of the SN in PD patients and other LB diseases.	(Wilhelmus et al. 2011) [78]
	LRP1	Increased LRP1 and ApoE in LB and melanised neurons of the SN in PD patients and other LB diseases.	[78]
	Aβ	Aβ-positivity is still relatively small in non-demented PD and MCI- PD (though prevalence may rise in increasing cognitive impairment and PDD).	(Mashima et al. 2017; Garon et al. 2021) [70,72]
		PD patients with enlarged Basal Ganglia PVS (BG-PVS) at baseline had reduced CSF Aβ42 and had lower Montreal cognitive assessment (MoCA) scores at 3-year follow-up compared to PD patients without enlarged BG-PVS	(Chen et al. 2022) [79]
		Increasing age was associated with increased cortical Florbetaben (an amyloid PET tracer) within PD patients.	(Melzer et al. 2019) [73]
		When age was adjusted, no correlation was found between cortical FBB uptake and global cognitive ability within the PD cohort.	(Melzer et al. 2019) [73]
		Aβ synergises with the other pathological processes, which can accelerate primary cognitive impairment seen in PD.	(Melzer et al. 2019; Garon et al. 2021) [70,73]

Abbreviations: Aβ = amyloid-β, ApoE = Apolipoprotein E, AQP4-/- = AQP4 deficient/knockout, AQP4+/+ = wild-type AQP4, LRP1 = Low-density lipoprotein receptor-related protein 1, MCI = mild cognitive impairment, MCI-PD = Parkinson’s Disease with mild cognitive impairment, MPTP = 1-methyl-4-phenyl-1,2,3,6-tetrahydropyridine, PD = Parkinson’s disease, PVS = perivascular space.

**Table 3 jcm-11-06964-t003:** Summary of the main findings from studies exploring glymphatic changes in Amyotrophic Lateral Sclerosis.

Cohort	Main Findings		References
ALS: Mouse models	AQP4 changes in the spinal cord	AQP4 expression increased in the spinal cord of SOD1G93A mice as the disease progressed	(Dai et al. 2017) [91]
		AQP4 polarization decreased as the disease progressed, and AQP4 polarized localization at the endfeet of astrocytes was decreased in the spinal ventral horn of SOD1G93A mice at the disease onset and end stages.	(Dai et al. 2017) [91]
	AQP4 changes in the BBB	Alternations to AQP4 in ALS may cause reduced BBB integrity, as a result, AQP4 changes lead to impaired potassium and connexin regulation, resulting in increased BBB permeability	(Cui et al. 2014; Zou et al. 2019) [92,93]
		SOD1G93A AQP4-/- ALS mouse models had improved BBB permeability compared to AQP4+/+ SOD1G93A mouse models, with reduced hemosiderin deposition and immunoglobulin leakage.	(Watanabe-Matsumoto et al. 2018) [99]
	Changes in the glymphatic system and disease outcomes due to AQP4 changes	Disease onset and lifespan were decreased significantly in AQP4^−/−^ *SOD1^G93A^* mouse models compared to AQP4^+/+^ *SOD1^G93A^* models	[99]

Abbreviations: AQP4-/- = AQP4 deficient/knockout, AQP4+/+ = wild-type AQP4, ALS = Amyotrophic Lateral Sclerosis, BBB = blood brain barrier, SOD1 = superoxide dismutase 1, SOD1G93A = SOD1 G93A mutation.

**Table 4 jcm-11-06964-t004:** Summary of the main findings from studies exploring glymphatic changes in Multiple Sclerosis.

Cohort	Main Findings		References
MS: Mouse models	AQP4 changes in demyelinated areas	In immunohistological analysis, cuprizone + EAE mice showed increased AQP4 expression at the centre of the inflammatory lesions with reduced AQP4 expression and polarity at the edge of the lesion	(Rohr et al. 2020) [114]
		In immunohistological analysis, Cuprizone toxin-induced demyelination mice models exhibited loss of AQP4 polarity within the endfeet of astrocytes surrounding perivascular structures at the perivascular endfeet of astrocytes and diffuse increase in AQP4 expression.	(Rohr et al. 2020) [114]
MS: Humans	Inflammatory changes in demyelinated areas	In immunohistological analysis, the post-mortem advanced MS cohort showed diffuse AQP4 expression increases in chronic-active lesions in advanced MS.	[114]
	MRI: DTI-ALPS	ALPS index was lower in both RRMS and progressive multiple sclerosis patients compared to HC, with progressive multiple sclerosis patients exhibiting lower ALPS values than RRMS patients	(Carotenuto et al. 2021) [118]
		Lower ALPS index score in the MS groups was associated with more severe clinical disability, more significant lesion load, grey matter (GM) atrophy, reduced mean diffusivity, and fractional anisotropy in normal-appearing white matter	(Carotenuto et al. 2021) [118]

Abbreviations: ALPS = along the perivascular space, AQP4-/- = AQP4 knockout, AQP4+/+ = wild-type AQP4, DTI = diffusion tensor imaging, DTI-ALPS = diffusion tensor imaging-along the perivascular space, EAE = Experimental autoimmune encephalitis, HC = healthy controls, MRI = magnetic resonance imaging, MRS = magnetic resonance spectroscopy, NAA = N-Acetyl-Aspartate, PD = Parkinson’s disease, PVS = perivascular space, RRMS = relapsing-remitting MS.

## Data Availability

Not applicable.

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
