# Peer review of "Research Evidence of the Role of the Glymphatic System and Its Potential Pharmacological Modulation in Neurodegenerative Diseases"

_jcm, 2022, doi:10.3390/jcm11236964_

Round 1

Reviewer 1 Report

The review of Verghese et al., focuses on the glymphatic system and its role in neurodegenerative diseases (NDs). The authors summarized all the relevant works to show the glymphatic system to be critical in several NDs including AD, PD, HD and ALS. Finally, the authors discuss the possibility to target the glymphatic system as a potential novel therapeutic approach in those uncurable diseases. 

I found this review well-written and interesting.

I do not have major comments.

However, I invite the authors to revise a few typo errors.

Moreover, I would like to suggest to reconsider the title, since this review largely describes studies on the role of the glymphatic system on NDs with a more marginal section regarding the pharmacological modulation.

Due to the limited number of data supporting the benefit of pharmacological modulation of the glymphatic system in NDs, any similar therapeutic approach should be considered accordingly and thus discussed in a more speculative way. 

Reviewer 2 Report

This is a well written review on the role of glymphatic system in neurodegenerative diseases. The narrative of this review is well balanced, and the authors describe general and specific information on the main topic. They focus on the role of aquaporin 4 in neurodegeneration and describe the pharmacological approaches that may modulate the glymphatic system in neurodegenerative diseases.

However, the connection between AQP4 and the mechanisms by which it can be pharmacologically regulated are not found in this review. How the modulation of AQP4 alters the levels of proteins associated to neurodegeneration?

Therefore, the authors must add an structural diagram with AQP4 that clearly describes how different molecules regulate this channel. The independent AQP4 pharmacological regulation of the glymphatic system must be also included in a diagram. Thus, the title of this review will be well justified.

Because AQP4 is the main focus of this review, it must be described in the abstract.

The available pdf of this manuscript do not contain the tables cited in the body text.

Round 2

Reviewer 2 Report

The authors made substantial changes to this review and answered my queries. Because they made several modifications I would suggest to proofread the manuscript again.